# The pathophysiological impact of intra-abdominal hypertension in pigs

**Robert Wise**[1,2,3]*, **Reitze Rodseth**[2], **Ester Párraga-Ros**[4], **Rafael Latorre**[4], **Octavio López Albors**[4], **Laura Correa-Martín**[5], **Francisco M. Sánchez-Margallo**[5], **Irma Eugenia Candanosa-Aranda**[6], **Jan Poelaert**[1], **Gregorio Castellanos**[7], **Manu L. N. G. Malbrain**[8,9,10]

1 Faculty Medicine and Pharmacy, Vrije Universiteit Brussel (VUB), Brussels, Belgium, 2 Discipline of Anaesthesiology, and Critical Care, Nelson R Mandela School of Medicine, University of KwaZulu-Natal, Durban, South Africa, 3 Adult Intensive Care Unit, John Radcliffe Hospital, Oxford University Hospitals Trust, Oxford, United Kingdom, 4 Department of Anatomy and Comparative Pathology, Veterinary Faculty, University of Murcia, Murcia, Spain, 5 Laparoscopy Department Jesus Uson Minimally Invasive Surgery Centre, Caceres, Spain, 6 Highlands Teaching and Research Farm, Faculty of Veterinary Medicine, National Autonomous University of Mexico, Queretaro. Mexico, 7 Department of General Surgery, Virgen de la Arrixaca General University Hospital, Murcia, Spain, 8 First Department of Anaesthesiology and Intensive Care Medicine, Medical University of Lublin, Lublin, Poland, 9 Medical Director (CMO), Medical Data Management, Medaman, Geel, Belgium, 10 International Fluid Academy, Lovenjoel, Belgium

* robwiseICU@gmail.com

**Data Availability Statement:** All relevant data are within the paper and its Supporting information files.

**Funding:** This work was supported by one grant from Extremadura Regional Government through

## Abstract

### Background

Intra-abdominal hypertension and abdominal compartment syndrome are common with clinically significant consequences. We investigated the pathophysiological effects of raised IAP as part of a more extensive exploratory animal study. The study design included both pneumoperitoneum and mechanical intestinal obstruction models.

### Methods

Forty-nine female swine were divided into six groups: a control group (Cr; n = 5), three pneumoperitoneum groups with IAPs of 20mmHg (Pn20; n = 10), 30mmHg (Pn30; n = 10), 40mmHg (Pn40; n = 10), and two mechanical intestinal occlusion groups with IAPs of 20mmHg (MIO20; n = 9) and 30mmHg (MIO30; n = 5).

### Results

There were significant changes (p<0.05) noted in all organ systems, most notably systolic blood pressure (SBP) (p<0.001), cardiac index (CI) (p = 0.003), stroke volume index (SVI) (p<0.001), mean pulmonary airway pressure (MPP) (p<0.001), compliance (p<0.001), pO2 (p = 0.003), bicarbonate (p = 0.041), hemoglobin (p = 0.012), lipase (p = 0.041), total bilirubin (p = 0.041), gastric pH (p<0.001), calculated glomerular filtration rate (GFR) (p<0.001), and urine output (p<0.001). SVV increased progressively as the IAP increased with no obvious changes in intravascular volume status. There were no significant differences between the models regarding their impact on cardiovascular, respiratory, renal and gastrointestinal systems. However, significant differences were noted between the two models at 30mmHg,

the Plan Regional de Investigación de Extremadura, Spain (PRI09A161 to Minimally Invasive Surgery Center Jesus Usón) to FSM and GCE. The funder had no role in study design, data collection, analysis, decision to publish, or preparation of the manuscript. The funders of the grant did not play any role in the study.

**Competing interests:** Robert Wise: Dr. Wise declares that he has no financial or personal relationships that may have inappropriately influenced him in writing this paper. He is currently a member of the WSACS. Reitze Rodseth: Prof. Rodseth declares that he has no financial or personal relationships that may have inappropriately influenced him in writing this paper. He is supported in part by the National Research Foundation of South Africa. Ester Párraga-Ros: Prof. Párraga-Ros declares that she has no financial or personal relationships that may have inappropriately influenced him in writing this paper. Rafael Latorre: Prof. Latorre declares that he has no financial or personal relationships that may have inappropriately influenced him in writing this paper. Octavio López Albors: Prof. Lopez Albors declares that he has no financial or personal relationships that may have inappropriately influenced him in writing this paper. Laura Correa-Martín: Prof. Correa-Martín declares that she has no financial or personal relationships that may have inappropriately influenced him in writing this paper. Francisco M. Sánchez-Margallo: Prof. Sánchez-Margallo declares that he has no financial or personal relationships that may have inappropriately influenced him in writing this paper. Irma Eugenia Candanosa-Aranda: Prof. Candanosa-Aranda declares that she has no financial or personal relationships that may have inappropriately influenced him in writing this paper. Jan Poelaert: Prof. Poelaert declares that he has no financial or personal relationships that may have inappropriately influenced him in writing this paper. Gregorio Castellanos: Prof. Castellanos declares that he has no financial or personal relationships that may have inappropriately influenced him in writing this paper. Manu L. N. G. Malbrain: MLNGM is Professor of Critical Care Research at the 1st Department of Anaesthesiology and Intensive Therapy, Medical University of Lublin, Poland. He is co-founder, past-President and current Treasurer of WSACS (The Abdominal Compartment Society, http://www.wsacs.org). He is member of the medical advisory Board of Pulsion Medical Systems (part of Getinge group, Solna, Sweden), Serenno Medical, Potrero Medical, Sentinel Medical Technologies and Baxter. He consults for BBraun, Becton Dickinson, ConvaTec, Spiegelberg, and Holtech Medical, and received

with MIO30 showing worse metabolic and hematological parameters, and Pn30 and Pn40 showing a more rapid rise in creatinine.

## Conclusions

This study identified and quantified the impact of intra-abdominal hypertension at different pressures on several organ systems and highlighted the significance of even short-lived elevations. Two models of intra-abdominal pressure were used, with a mechanical obstruction model showing more rapid changes in metabolic and haematological changes. These may represent different underlying cellular and vascular pathophysiological processes, but this remains unclear.

## Introduction

Intra-abdominal hypertension (IAH) is present in 25–30% of critically ill patients on admission and causes profound systemic physiological derangement [1–10]. The consequences include renal dysfunction, prolonged intensive care and hospital stays, increased morbidity, multi-organ failure, and mortality [11]. Increased pressure within the intra-abdominal compartment reduces abdominal perfusion pressure (calculated as mean arterial pressure minus intra-abdominal pressure (IAP)), [12] thereby impairing perfusion to the intra- and extra-abdominal organs and, in particular, watershed areas such as the intestinal mucosa. Venous compression increases venous pressure resulting in congestion, intestinal edema, and gastrointestinal bacterial translocation [13].

The interaction between different organ systems within neighbouring anatomical compartments has been described as polycompartment syndrome [14, 15]. It is for this reason we have studied multiple organ systems together. This study includes two models of raised intraabdominal pressure, a pneumoperitoneum and a mechanical intestinal obstruction model [16]. We have previously demonstrated that intestinal vascular occlusion may differ between a pneumoperitoneum versus a mechanically obstructed model [16]. This may affect organ systems differently. Lactate increased substantially in a mechanically obstructed model which may reflect increased anaerobic metabolism because of imbalances in cardiorespiratory dynamics. This may in turn promote an inflammatory state but has yet to be proven, and probably requires a longer period of sustained elevated IAP.

The primary aim was to study the systemic effects of raised IAP on various organ systems. We used both the pneumoperitoneum and mechanical obstruction models to generate elevated intra-abdominal pressure. Data from all pigs, regardless of the model used, were included in the analysis for the primary outcome. We felt it important to study these effects across various IAP levels, as this has not been well documented previously. The secondary aim was to describe the effects between the two different models. We hypothesised that there would be no conceivable difference between the two models because of the relatively short duration of raised IAP.

## Methods

### Regulatory issues

The study was performed in accordance with the recommendations in the Royal Decree 1201/2005 of 10 October 2005 on the protection of animals used for experimentation and other scientific purposes. All experimental protocols were approved by the Committee on the Ethics of

speaker's fees from PeerVoice. He holds stock options for Serenno and Potrero. He is co-founder and President of the International Fluid Academy (IFA). The IFA (http://www.fluidacademy.org) is integrated within the not-for-profit charitable organization iMERiT, International Medical Education and Research Initiative, under Belgian law.

Animal Experiments of Minimally Invasive Surgery Centre Jesús Usón and by the Council of Agriculture and Rural Development of the Regional Government of Extremadura (No. ES100370001499), Spain.

## Animal study population

Forty-nine white female pigs (24.1 kg; range 17.3–33 kg) were fasted for 24 hours before receiving premedication with intramuscular atropine (0.04 mg/kg), diazepam (0.4 mg/kg) and ketamine (10 mg/kg). Induction and anesthesia were the same as described previously by Correa-Martin et al. [16]. The animals were pre-oxygenated with fractional inspired oxygen of 1.0 (fresh gas flow of 3–5 l/min), before propofol 1% (3 mg/kg) was administered. They were intubated, mechanically ventilated, and anesthesia was maintained with isoflurane (MAC of 1.25). Intravenous 0.9% sodium chloride fluid (2 ml/kg/h) and a remifentanil infusion (0.3 mcg/kg/min) for analgesia was provided intraoperatively. The animals were euthanised using potassium chloride (1–2 mmol/kg) on completion of the study, as per the American Veterinary Medical Association Panel on Euthanasia guidelines.

## Study design

The swine were consecutively allocated into six groups: a single control group (Cr; n = 5), three pneumoperitoneum groups with IAPs of 20 mmHg (Pn20; n = 10), 30 mmHg (Pn30; n = 10), and 40 mmHg (Pn40; n = 10), and two mechanical intestinal occlusion groups with IAPs of 20 mmHg (MIO20; n = 9) and 30 mmHg (MIO30; n = 5). Each group was studied for three and five hours, except the MIO30 that was only studied for 3 hours. A laparoscopic insufflation technique was used for the pneumoperitoneum model. The mechanical intestinal obstruction model, which was previously published, was achieved by placing a laparoscopic suture at the ileocaecal valve and infusing 0.9% saline into the bowel (16).

All the swine were positioned supine. IAP was measured using three methods simultaneously, with the direct intraperitoneal measurement technique used as the method to achieve the IAP target [16]. Gastric and bladder pressures were measured using the same methods as those described for humans in the WSACS consensus guidelines [16–18]. Multiple physiological parameters, together with blood samples, were measured every 30 minutes. Once IAP was stabilised, measurements were initiated and designated as T1. The control group did not have any intervention to increase IAP and received the same anesthetic and 30-minute physiological measurements as the experimental groups.

## Data collection

IAP was measured simultaneously at 30 minutes intervals in each pig using the three methods (i.e., transperitoneal [TP], transvesical [TV], and transgastric [TG]). The direct TP technique was considered the gold standard as it was a direct measure of IAP. A Jackson-Pratt catheter was inserted laparoscopically into the abdominal cavity and placed on the liver to perform TP IAP measurements [16]. A Foley catheter in the bladder and urine drainage bag were used for TV IAP measurements together with a manual manometer system (Holtech Medical, Charlottenlund, Denmark). An endoscopically placed gastric balloon-tipped catheter was connected to an electronic pressure transducer (Spiegelberg, Hamburg, Germany) to measure TG IAP. Transgastric measurements were graphically recorded in real time. The results of these comparisons have been previously reported [19].

Physiological signs and laboratory tests were collected every 30 minutes. Cardiovascular parameters measured included heart rate, blood pressure, cardiac output parameters (cardiac output, pulse pressure variation (PPV), stroke volume variation (SVV), assessed with

continuous electrocardiogram, pulse oximetry (with hemodynamic E-modulus PRESTN anesthesia monitor S/5TM General Electric Datex-Ohmeda®), invasive arterial blood pressure measurement and pulse contour cardiac output (PiCCO, Getinge, Sölna, Sweden) device. Respiratory parameters measured included respiratory rate, arterial oxygen saturation ($SpO_2$), partial pressure of carbon dioxide ($pCO_2$), mean pulmonary artery pressure (MPP), and ventilation parameters (tidal volumes (Vt), compliance, plateau pressure (PP), minute volume). These measurements came from the anesthetic monitor, ventilator, and arterial blood gases. Ventilation was provided with an $FiO_2$ of 1.0, a tidal volume of 10-15ml/kg, positive end-expiratory pressure of $2cmH_2O$, at a rate set (14 breaths/minute) to obtain an end-expiratory $CO_2$ partial pressure of 35-40mmHg. Metabolic and hematological parameters from blood samples included pH, base excess, bicarbonate, lactate, APTT, INR, and platelets. Gastrointestinal parameters from blood samples included ALT, ALP, GGT, LDH, total bilirubin, and lipase. An endoscopically inserted monitor connected to the anesthesia monitor (S/5TM General Electric Datex-Ohmeda, Helsinki, Finland) measured continuous gastric mucosal pH via tonometry. Renal function was assessed via a calculated glomerular filtration, urine output, and blood samples for urea and creatinine.

## Statistical analysis

We performed a statistical power analysis for sample size estimation from data analysing hemodynamic parameters from previous pig studies with an alpha of 0.05 and power equal to 0.80 [20–22]. With an alpha of 0.05 and power of 80%, the projected sample size needed with this effect size for the primary endpoints was 4 (Appendix A in S2 File supplementary electronic media). The proposed sample size, both in groups and combined by model, was adequate for the objectives of this study.

Categorical variables were described as proportions (%) and compared using Chi-square or Fisher exact test. Continuous variables are presented as means with standard deviations when normally distributed, and as medians with interquartile ranges when non-normal distribution occurred. Normality was evaluated with the Kolmogorov-Smirnov test. One-way ANOVA or Kruskal-Wallis tests were used to compare continuous variables as appropriate.

Physiological parameters were clustered into six groups: cardiovascular, respiratory, metabolic, hematological, gastrointestinal, and renal. These parameters were considered dependent variables. The interdependencies between the physiological parameters were explored for completeness. A linear mixed model was used to estimate changes in the physiological parameters over time, with measurements every 30 minutes. The mixed model accounted for the dependencies over time and the imbalanced nature of the data with a random intercept for the pig-specific averages and a random slope for the pig-specific changes over time. Pig-specific variances were also allowed. Not all combinations of treatment and pressure were present in the dataset. Therefore, specific comparisons were created to analyse the appropriate subgroups of observations. Various variables were highly correlated. Some were on an interval scale, while others were binary.

As a result, a combination of Pearson, polyserial, and polychoric correlations was performed. Data evolution over time was plotted to represent the relationship between pressure, IAP model, and duration. To assist in interpretation, all data were rescaled by a factor of 10. Rank 10 refers to subtracting each time point by ten so that ten becomes zero, and five becomes minus five. The advantage is that the value of the intercept can be interpreted. This value is the expected value if all predictors in the model were zero, and this is now true for a time equal to ten.

A linear mixed model was used to analyse the physiological parameters to explain the observed scores while incorporating changes over time. We used the Shaffer p-value correction for multiple comparisons. All analyses were performed using R version 3.5.3 (R Core Team (2017). R: A language and environment for statistical computing. R Foundation for Statistical Computing, Vienna, Austria. URL https://www.R-project.org/).

This is a secondary, but more complete analysis, of a study previously published by the same group (16).

## Results

The pathophysiological effects of elevated intra-abdominal pressure on various organ systems are shown in the following tables: cardiovascular (Table 1), respiratory (Table 2), metabolic and hematological (Table 3), gastrointestinal (Table 4), and renal systems (Table 5).

### 1. Cardiovascular system

**Blood pressure.**   There was a decrease in both systolic and diastolic pressure as IAP increased and as time progressed. The most significant changes occurred at the highest pressures. A strong intra-class correlation was present.

For **systolic blood pressure**, the downward evolution occurred in all models but was more substantial at higher pressures. **Diastolic pressure** changes and relationships were similar to systolic pressure changes, only less significant (S3 Fig in S2 File).

**Cardiac output parameters.**   IAH lowered the cardiac output and stroke volume index in all models (S4 Fig in S2 File). The most significant changes occurred in the MIO models.

**Stroke volume index** decreased significantly (61.1%), with the greatest changes seen as the pressure increased. **Stroke volume variation** (SVV) and **pulse pressure variation** (PPV) increased (100.9% and 93.5% respectively, p<0.001) across all groups and in both models compared to the control group.

### 2. Respiratory system

**Mean pulmonary airway pressure (MPP).**   MPP increased (62.5%, p<0.001) across all group, with the highest pressures corresponding to the highest IAP. There were almost no differences for MPP between the groups except for Pn30 and Pn40, which differed significantly (p<0.001) (S7 Fig in S2 File). There seemed to be an inverse relationship between MPP and weight.

**Table 1. Cardiovascular parameters regardless of IAP model presented as median values with interquartile ranges and significance.**

|  | Control | IAP 20mmHg | IAP 30mmHg | IAP 40mmHg | p-value |
|---|---|---|---|---|---|
|  | n = 5 | n = 19 | n = 15 | n = 10 |  |
| Heart rate (beats/min) | 104 (14) | 92 (41.0) | 113.0 (42.5) | 117 (58.5) | <0.001 |
| Systolic BP (mmHg) | 86.0 (17.0) | 57.0 (20.0) | 61.0 (15.0) | 64.0 (16.0) | <0.007 |
| Cardiac index (L/min/m$^2$) | 4.2 (1.3) | 2.03 (0.86) | 2.28 (0.83) | 2.31 (0.82) | <0.001 |
| SVI (mL/m$^2$) | 39.0 (13.2) | 21.0 (6.8) | 20.0 (6.0) | 17.0 (9.5) | <0.001 |
| PPV (%) | 17.0 (5.8) | 26.0 (7.0) | 25.0 (7.0) | 28.0 (3.0) | <0.001 |
| SVV (%) | 19.5 (7.0) | 26.0 (8.0) | 30.0 (7.0) | 30.0 (5.0) | <0.001 |

BP = blood pressure; SVI = stroke volume index; PPV = pulse pressure variation; SVV = stroke volume variation

**Table 2. Respiratory parameters regardless of IAP model, presented as median values with interquartile ranges and significance.**

| | Control | IAP 20mmHg | IAP 30mmHg | IAP 40mmHg | p-value |
|---|---|---|---|---|---|
| | n = 5 | n = 19 | n = 15 | n = 10 | |
| MPP (mmHg) | 5.0 (0.0) | 6.0 (0.0) | 7.0 (1.0) | 8.0 (1.8) | <0.001 |
| Compliance (mL/cmH$_2$0) | 18.0 (2.5) | 8.4 (1.9) | 6.3 (2.0) | 6.5 (1.4) | <0.001 |
| pO$_2$ (mmHg) | 355.5 (70.1) | 407.0 (113.0) | 489.0 (187.3) | 371.0 (83.0) | 0.003 |
| VT$_E$ (ml) | 290.0 (75.0) | 210.0 (50.0) | 240.0 (50.0) | 275.0 (40.0) | <0.002 |
| pCO$_2$ (mmHg) | 42.2 (5.0) | 47.8 (17.3) | 46.1 (11.5) | 37.0 (11.6) | 0.091 |
| Plateau pressure (cmH$_2$0) | 18.0 (3.0) | 27.0 (6.0) | 34.0 (6.8) | 43.0 (10.0) | <0.001 |
| Driving pressure (cmH$_2$0) | 17.0 (3.0) | 26.0 (5.0) | 32.0 (7.0) | 42.0 (11.0) | <0.001 |

MPP = mean pulmonary airway pressure; pO$_2$ = arterial partial pressure of oxygen; VT$_E$ = expiratory minute volume; pCO$_2$ = arterial partial pressure of carbon dioxide

**Compliance.** Dynamic compliance [calculated as tidal volume divided by (plateau pressure minus PEEP)] was reduced (60.2%, p<0.001) in all IAH models, with moderate intra-class correlation (S8 Fig in S2 File).

**Partial pressure of oxygen (pO$_2$).** The partial pressure of oxygen differed in all the IAH models compared to the control. A strong intra-class correlation was present.

**Expiratory minute volume (MVE).** Expiratory minute volume was decreased (33.6%, p<0.002) for all models compared to the control group (S10 Fig in S2 File).

**Partial pressure of carbon dioxide (pCO$_2$).** None of the comparisons suggested any difference in pC0$_2$ readings. Lower pCO$_2$ readings were seen in heavier pigs (S11 Fig in S2 File).

**Plateau pressure (PP) and driving pressure.** Driving pressure and PP (125.6%, p<0.001) increased with IAP, with the highest values being reached in the P40 and MIO30 groups (Fig 1). Especially noticeable were some extreme increases in the MIO30 and Pn40 groups (p<0.001).

## 3. Metabolic and hematological systems

**Base excess (BE).** The base excess decreased (102.1%, p = 0.088) in nearly all models compared to the control group (S14 Fig in S2 File). The Pn20 group differed the least when compared to the control.

**Bicarbonate (HCO$_3$).** Bicarbonate responded similarly to base excess, with readings decreasing (12.5%, p = 0.038) more rapidly in all groups when compared to the control group (except in the Pn20 group) (S15 Fig in S2 File).

**Table 3. Metabolic and hematological parameters regardless of IAP model, presented as median values with interquartile ranges and significance.**

| | Control | IAP 20mmHg | IAP 30mmHg | IAP 40mmHg | p-value |
|---|---|---|---|---|---|
| | n = 5 | n = 19 | n = 15 | n = 10 | |
| Base excess (mmol/L) | 4.0 (3.0) | 2.6 (7.0) | -1.0 (8.3) | -6.0 (8.5) | 0.088 |
| Bicarbonate (mmol/L) | 28.5 (4.5) | 27.7 (7.1) | 24.7 (7.3) | 20.1 (8.6) | 0.038 |
| Haemoglobin (g/dL) | 7.6 (0.8) | 8.6 (0.8) | 9.3 (1.5) | 10.0 (1.5) | 0.021 |
| Platelets (x10$^9$/L) | 412.0 (140.0) | 429.0 (144.0) | 436.0 (248.8) | 397.0 (98.5) | <0.001 |
| APTT (sec) | 14.3 (6.2) | 24.0 (40.0) | 29.0 (46.4) | 55.0 (65.3) | 0.124 |
| INR | 0.8 (0.2) | 1.2 (0.21) | 1.14 (0.30) | 1.07 (0.24) | 0.652 |

APTT = activated partial thromboplastin time; INR = international normalised ratio

**Table 4. Gastrointestinal parameters regardless of IAP model, presented as median values with interquartile ranges and significance.**

| | Control | IAP 20mmHg | IAP 30mmHg | IAP 40mmHg | p-value |
|---|---|---|---|---|---|
| | n = 5 | n = 19 | n = 15 | n = 10 | |
| LDH (U/L) | 1046.0 (153.5) | 903.0 (406.0) | 1100.0 (338.0) | 1032.0 (349.0) | 0.834 |
| Lipase (U/L) | 3.6 (0.8) | 6.5 (1.9) | 8.7 (5.95) | 8.5 (4.5) | 0.032 |
| ALT (IU/L) | 38.5 (7.8) | 30.0 (10.3) | 37.5 (17.4) | 27.0 (13.5) | 0.001 |
| GGT (IU/L) | 30.0 (13.5) | 30.0 (10.5) | 34.0 (16.0) | 35.0 (12.0) | 0.798 |
| ALP (U/L) | 368.5 (217.5) | 339.0 (138.0) | 321.0 (93.0) | 383.9 (163.8) | 0.023 |
| Total bilirubin (mg/dL) | 0.3 (0.2) | 0.15 (0.14) | 0.30 (0.18) | 0.42 (0.38) | 0.037 |
| Gastric tonometry (pH) | 7.24 (0.06) | 7.13 (0.14) | 6.91 (0.28) | 6.68 (0.37) | <0.001 |
| pCO$_2$ gap (mmHg) | 3.00 (1.36) | 11.20 (12.72) | 11.40 (17.84) | 25.0 (10.0) | <0.001 |

LDH = lactate dehydrogenase; ALT = alanine aminotransferase; GGT = Gamma-glutamyl transferase; ALP = alkaline phosphatase

**Hematology.** Hematological parameters mainly showed a positive correlation in both models and at all pressures. Platelet levels revealed an increasing trend from lower IAPs to the higher IAPs in both models (25.7% increase regardless of model) and were significant (p<0.001). Activated partial thromboplastin time (APTT) had an upwards trend (517.3% increase regardless of model, p = 0.124) and was significant in the MIO group. The international normalised ratio (INR) was relatively consistent across both models and at all pressures.

## 4. Gastrointestinal system

**Lactate dehydrogenase (LDH).** LDH increased (14.4% increase, p = 0.834) as IAP increased, however, the changes were not significant. The Pn20 model increased more rapidly than in the control group (S17 Fig in S2 File). There were outliers in each group. The pigs with the highest LDH showed higher mortality.

**Lipase.** The increase in lipase (147.3%, p = 0.032) was seen across all groups, with the higher values corresponding to higher IAP. Some pigs in the PN30 and Pn40 groups showed the greatest increases in lipase.

**Liver tests.** Alanine aminotransferase (ALT) increased over the 20mmHg and 30mmHg groups, regardless of the pressures. ALT unexpectantly decreased in the PN40 group.

**Gamma-glutamyl transferase (GGT)** decreased (6.9% decrease, p = 0.798) and showed a difference in evolution between Pn20 and Pn30. There was also a difference between the control and each of the 20mmHg conditions (p = 0.038 and 0.044 respectively). There were significant differences observed for all groups at rank 10 (p<0.001).

**Table 5. Renal parameters regardless of IAP model, presented as median values with interquartile ranges and significance.**

| | Control | IAP 20mmHg | IAP 30mmHg | IAP 40mmHg | p-value |
|---|---|---|---|---|---|
| | n = 5 | n = 19 | n = 15 | n = 10 | |
| Calculated GF (mmHg) | 59.0 (17.5) | 10.0 (24.0) | -14.5 (18.5) | -34.0 (15.0) | <0.001 |
| Urine output (ml/kg/hr) | 2.8 (1.0) | 0.39 (0.95) | 0.36 (0.15) | 0.23 (0.18) | <0.001 |
| Urea (mg/dL) | 20.7 (9.4) | 24.5 (11.1) | 25.1 (9.1) | 26.3 (9.2) | 0.349 |
| Creatinine (mg/dL) | 2.3 (0.6) | 2.69 (1.14) | 2.73 (0.79) | 2.33 (0.44) | 0.283 |

GF = glomerular filtration

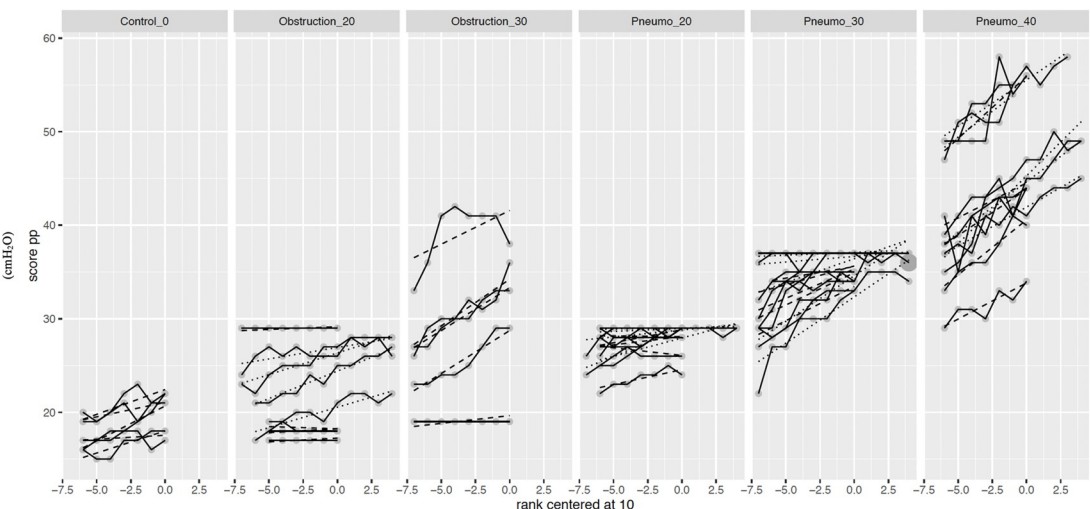

**Fig 1. Scatter plot of plateau pressure variables.**

**Alkaline phosphatase (ALP)** and **total bilirubin (totBil)** both showed gradual, yet significant changes (31.2% decrease, p = 0.023 and 50% increase, p = 0.037, respectively) as IAP rose. There were significant differences observed for all groups at rank 10 for ALP (p<0.001).

**Gastric tonometry.** There was a significant and progressive decrease in **pH** for **gastric tonometry** as the intra-abdominal pressure increased. The decline was this most marked when the IAP reached 40mmHg (a 4% reduction). The $pCO_2$ gap (a surrogate marker for cardiac output) increased significantly as the IAP rose.

## 5. Renal system

**Calculated glomerular filtration.** Glomerular filtration was calculated using the formula: filtration gradient = MAP–(2 x IAP). IAP measures were obtained transvesically, transperitoneally, and transgastrically. As time progressed, there was a downward trend for all groups (104.6%, p<0.001). The most significant changes were seen with the highest pressures (Fig 2). A strong intra-class correlation was noted.

**Urine output.** Urine output was significantly greater in the control group when compared to each of the groups. Urine output decreased with the increasing IAP (S20 Fig in S2 File). At worst, this reduction was 74.6% (p<0.001).

**Urea.** Urea increased (20.1%, p = 0.349) in the IAH models compared to the control (S21 Fig in S2 File). Moderate intra-class correlation was identified.

**Creatinine.** Creatinine increased (5.7%, p = 0.283) as time progressed in all IAH models compared to the control group (S22 Fig in S2 File). There was a strong intra-class correlation.

**Secondary objectives (refer to supplementary data).** The comparison between the pneumoperitoneum model and the mechanical obstruction model did not show any significant differences in the **cardiovascular** or **respiratory** parameters measured. The changes were at times more rapid when the abdominal pressures were higher. There were minor differences observed between groups.

**Metabolic** parameters showed a decreased base excess across both models. The Pn20 group differed the least when compared to the control. Bicarbonate decreased in both models but was more marked in the MIO model compared to the PN group at 20mmHg (S15 Fig in S2

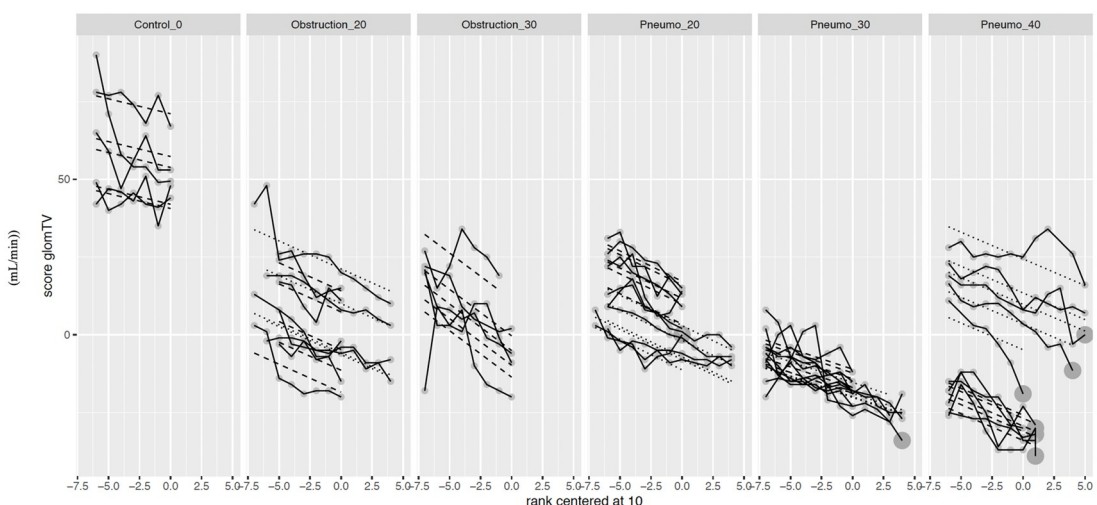

**Fig 2. Scatter plot of calculated glomerular filtration variables.**

File). **Haematological** changes were consistent across the models and groups, however, the MIO30 group showed greater changes in INR, platelets and APTT.

The **gastrointestinal system** revealed unexpected results. The **bilirubin and ALT** decreased or only marginally increased in both 20mmHg models. Bilirubin rose the most when pressures increased, especially at 40mmHg. There were no significant differences between models for GGT and ALP.

The evolution of increase for **LDH** was greater in the Pn compared to the MIO models (S17 Fig in S2 File).

The opposite occurred for **lipase**, where lipase increased across both models and at all pressures (S18 Fig in S2 File). The evolution of increase for **lipase** and **ALT** were strongest in the MIO30 group.

**Renal** parameters reflected similar trends independent of the model of intra-abdominal pressure used. The creatinine increased more sharply in the pneumoperitoneum groups, especially Pn30 and Pn40 (S22 Fig in S2 File).

## Discussion

Several previous studies aimed to investigate aspects of the pathophysiological consequences of IAP. This study found worsening cardiorespiratory function and a predictable effect on renal function. The metabolic, hematological, and gastrointestinal effects are significant for some, but not all, parameters studied. This may be a result of the short duration of the implemented models, with insufficient time to reach significance. Alternatively, it may represent more complicated pathophysiology or protective mechanisms in early IAH. Metabolic changes showed a positive correlation in the Pn model, but a negative correlation in the MIO group. The reason for this is uncertain but may represent different underlying pathophysiology related to the effects of mechanical obstruction. The results are discussed according to organ system.

### Cardiovascular system

The cardiovascular changes observed were as expected following the rise in IAP with a subsequent increased afterload, a decreased preload, and thereby decreased cardiac output.

Additionally, as the stroke volume is determined by a combination of preload, contractility, afterload, and heart rate, there were significant increases in heart rate to compensate for a reduced stroke volume (heart rate increased by 40.2% regardless of model). This was particularly noticeable when IAP was 30mmHg or more.

An important finding was the relationship between elevated intra-abdominal pressure and markers of fluid responsiveness. SVV and PPV have been helpful in volume status assessment; however, the impact of intra-abdominal pressure appears to make these measurements difficult to interpret [21, 23]. Previous research has published conflicting results. Deloya et al. found an IAP greater than 15mmHg to cause significant changes in SVV despite the subjects being normovolaemic.

Contrary to this, Jacques et al. concluded that SVV was still helpful in the face of elevated IAP, but different thresholds may apply. These two previous studies were pig models; however, Liu et al. examined the same question in patients undergoing laparoscopic cholecystectomy [23]. The findings showed SVV increased progressively as the IAP was raised with no changes in volume status. In our study, the pigs were presumed to have been kept euvolemic with a constant infusion of IV crystalloids and only insensible losses. Despite this, IAP lowered the cardiac output in all models and increased PPV and SVV across all groups and in both models. This supports previous findings of the difficulty in using cardiovascular markers of volume status that use respiratory variation in subjects with elevated IAP. Measurement of IAP should be considered when using SVV or PPV, and for the utility of SVV should be determined in patients with IAP.

Interestingly, the systolic BP and cardiac index were closer to the control group values in the animals with higher IAP (40mmHg) versus lower IAP (20mmHg). Theoretically, these haemodynamic findings may be the consequence of the mechanical and neurohormonal responses to extremely elevated intra-abdominal pressure. These are related to inferior vena cava compression, aortic compression, decreased splanchnic blood flow, decreased renal blood flow, and diaphragmatic displacement. The overall result of these effects includes increased right atrial pressure, increased systemic vascular resistance, countered by a possible decrease in cardiac output. However, a slightly elevated systolic BP may be seen with a higher cardiac index at higher IAP if the effects on SVR and RAP are greater than the effects on cardiac output. Another explanation might be the mathematical coupling between high levels of IAP and AP due to the mechanical transmission of pressures from one compartment to another as is seen in the polycompartment syndrome.

## Respiratory system

The pressure exerted on the diaphragm by high IAP causes decreased tidal volumes, compliance, functional residual capacity, and increased respiratory and airway pressures [24–29]. Previous animal and human studies have demonstrated a non-uniform decrease in lung volume and functional residual capacity with abdominal distension and elevated IAP. Most studies are conducted over hours, and hypothetically, we expect these findings to worsen as time progresses. Both chest wall and lung compliance are affected. The progressive decrease in compliance across both models and at each IAH pressure, with significant negative correlation, was in keeping with previous studies [26, 28, 30]. MPP and plateau pressure) measurements increased with a significant positive correlation in most groups, the exception being the MIO20 group. IAH appears to increase both inspiratory airway and pleural pressures, and thus does not seem to affect transpulmonary pressures significantly.

The $pO_2$ decreased in the Pn40 model, but the apparent increase in $pO_2$ in the other models requires further exploration. Ventilation-perfusion relationships have been studied in animals

and humans [27, 31–33]. There appears to be a redistribution of blood flow from atelectatic and dependent areas of the lung when IAP is raised. This causes a redistribution of blood flow to improve the ventilation-perfusion mismatch. This compensatory mechanism may account for the increases in $pO_2$ in some groups. These changes may occur rapidly, but how and when it is overwhelmed requires further research. The partial pressure of carbon dioxide responded as predicted.

PEEP was not adjusted during the study despite increasing IAP. This was done to assess the effects of the rising IAP. Previous animal studies have investigated the effects of PEEP and demonstrated an improvement in end-expiratory lung volumes, shunt fraction, dead space, and oxygenation when matching PEEP levels with IAP. This study did not intend to describe the effects of PEEP [27, 28, 34]. The importance of recognising increased IAP and its potential influence on the chest wall and lung compliance, tidal volumes, and functional residual capacity, should be a focus of ventilation teaching. Incorporation of IAP measurement into ventilator setting decision pathways should be considered.

### Metabolic and hematological systems

Interestingly, the metabolic findings showed varying results between the models, but overall, values decreased in nearly all models compared to the control group. These changes were relatively rapid for base excess and serum bicarbonate, and highlight the importance of early identification of IAH.

The increasing platelet trend was unexpected. Mean platelet volume (MPV) has been studied in response to elevated IAP in patients undergoing laparoscopic cholecystectomy as a model for IAH [35–41]. Interestingly, there was a significant increase in MPV on insufflation and a decrease with deflation, confirming a platelet response to elevated IAP. There are several theories about platelet response to elevated IAP. MPV has previously been linked as a marker of inflammation to several different pathologies, including septicemia, cardiac, respiratory and intra-abdominal pathologies [42–44]. The platelet response may be driven by the secretion of free oxygen radicals and inflammatory mediators in response to poor abdominal perfusion. This in turn promotes the production of platelets through the stimulation of megakaryocytes [45]. Other theories suggest hormonal action on megakaryocytes, the retention of small platelets in ischaemic organs, and platelet swelling in response to thrombopoietin [46–48]. However, the increase in platelets over a relatively short time (maximum 5 hours), despite the long average lifespan of platelets (8–10 days), suggests a relatively rapid process that requires further investigation.

### Gastrointestinal system

In this study, GGT, alkaline phosphatase, and LDH all showed progressive change as the IAP increased in both models. There was a progressive increase in the lipase measurements, most marked in the Pn and MIO30 groups, with corresponding statistical significance. Total bilirubin increased significantly between the control group and IAP 40mmHg.

The $pCO_2$ ($pCO_2$ gap = $PcvCO_2$ – $PaCO_2$) has previously been identified as a surrogate marker for cardiac output and used in various settings [49–53]. A $pCO_2$ gap >6mmHg suggests a persistent shock state. All groups of raised IAP had a $pCO_2$ gap of >6mmHg and rose significantly with the IAP increase.

### Renal system

Renal dysfunction occurs with elevated intra-abdominal pressure because of venous congestion, increased left ventricular afterload and decreased renal perfusion, and a systemic

inflammatory response [54–57]. The compression of renal veins will result in venous congestion and a decrease in glomerular filtration with an increase in renin release. Diminished renal perfusion and salt and water retention may contribute to a vicious cycle of fluid accumulation and increasing IAP [58, 59].

Creatinine, urea, glomerular filtration and urine output all demonstrated changes that were expected as a consequence of the pathophysiology described above. The lack of statistically significant changes, for all but glomerular filtration, may be due to the short duration of the raised IAP.

### Comparison between raised IAP models

The lack of differences between the two models in the cardiorespiratory parameters may be expected when both models had the same IAP, and the inflammatory responses may not have had enough time to influence contractility through inflammatory mediated negative inotropy.

The metabolic effects of IAP appear to be influenced by the mechanism of intra-abdominal hypertension [16]. The explanation may be linked to the impact on bowel mucosa and underlying tissue that was ligated as part of the model (like that which occurs in bowel obstruction). This may result in a perfusion deficit, as shown by the decrease in pHi.

Lipase appears to be a useful marker to monitor, regardless of the model of IAH. In comparison, total bilirubin increased significantly between the control group and IAP 40mmHg but did not reveal any trends or consistent statistically significant correlation between the two models.

Glomerular filtration was shown to be a useful marker of renal decline and showed a statistically significant correlation in both models and at all pressures (Pn20 vs. MIO20 and Pn30 vs. MIO30).

### Limitations

Although care was taken to control and standardise all interventions, there may have been a difference in each animal's underlying physiology. While this was a relatively large animal study, there were only five or fewer animals in each group. Due to funding limitations, one group had four, which still satisfied our sample size requirements. The implications of this include a limited ability to identify statistically significant changes, especially when parameters are measured over only a few hours. Set pressures were used versus gradually increasing pressures over time which may have been closer to clinical situations. However, we chose this design to define the size of the groups, as pigs may not have survived raised pressures for long periods, thus missing out on observing what happens at 30mmHg and 40mmHg. We did not measure readings beyond 5 hours, and there was not a 40mmHg intervention for the mechanical obstruction group. Consequently, changes that take longer to manifest, such as elevation of creatinine and urea, would not be identified. The effect of PEEP could not be removed completely because a PEEP of zero was not possible on the anaesthetic machine used. Unfortunately, lactate readings, global end-diastolic volume, and extravascular lung water could not be reported because of incomplete data.

### Conclusion

This study identified and quantified the impact of intra-abdominal hypertension at different pressures on several organ systems and highlighted the significance of even short-lived elevations. Two models of intra-abdominal pressure were used, with a mechanical obstruction model showing more rapid changes in metabolic and haematological changes. These may

represent different underlying cellular and vascular pathophysiological processes, but this remains unclear.

## Supporting information

**S1 Data.**
(PDF)

**S1 File.**
(PDF)

**S2 File. Electronic supplementary media.** The file contains S1-S22 Figs, S1, S2 Tables, and Appendix A.
(PDF)

## Acknowledgments

We would like to acknowledge Wilfried Cools from the Interfaculty Center Data processing and Statistics at the Vrije Universiteit Brussel (VUB), for his statistical analysis contribution.

## Author Contributions

**Conceptualization:** Gregorio Castellanos.

**Formal analysis:** Robert Wise, Reitze Rodseth, Jan Poelaert, Manu L. N. G. Malbrain.

**Funding acquisition:** Gregorio Castellanos.

**Investigation:** Ester Párraga-Ros, Rafael Latorre, Octavio López Albors, Laura Correa-Martín, Francisco M. Sánchez-Margallo, Irma Eugenia Candanosa-Aranda, Gregorio Castellanos.

**Methodology:** Ester Párraga-Ros, Rafael Latorre, Octavio López Albors, Laura Correa-Martín, Francisco M. Sánchez-Margallo, Irma Eugenia Candanosa-Aranda, Gregorio Castellanos.

**Validation:** Robert Wise, Reitze Rodseth, Jan Poelaert, Manu L. N. G. Malbrain.

**Writing – original draft:** Robert Wise, Reitze Rodseth, Jan Poelaert, Manu L. N. G. Malbrain.

**Writing – review & editing:** Robert Wise, Reitze Rodseth, Ester Párraga-Ros, Rafael Latorre, Octavio López Albors, Laura Correa-Martín, Francisco M. Sánchez-Margallo, Irma Eugenia Candanosa-Aranda, Jan Poelaert, Gregorio Castellanos, Manu L. N. G. Malbrain.

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
