## [Decision Letter · Decision Letter 0]

31 Jan 2023

PONE-D-22-32095Pathophysiological impact of different models of intra-abdominal hypertension in pigsPLOS ONE

Dear Dr. Wise,

Thank you for submitting your manuscript to PLOS ONE. After careful consideration, we feel that it has merit but does not fully meet PLOS ONE’s publication criteria as it currently stands. Therefore, we invite you to submit a revised version of the manuscript that addresses the points raised during the review process.

Your work on experimental IAH is very comprehensive and provides a lot of data. Nevertheless, your manuscript should be revised for publication. Regarding the revision, I refer to the reviews listed below.

We look forward to receiving your revised manuscript.

Kind regards,

Alexander Wolf

Academic Editor

PLOS ONE

Journal Requirements:

Reviewers' comments:

Reviewer's Responses to Questions

**Comments to the Author**

1. Is the manuscript technically sound, and do the data support the conclusions?

Reviewer #1: Partly

Reviewer #2: Partly

2. Has the statistical analysis been performed appropriately and rigorously? 

Reviewer #1: I Don't Know

Reviewer #2: Yes

3. Have the authors made all data underlying the findings in their manuscript fully available?

Reviewer #1: Yes

Reviewer #2: Yes

4. Is the manuscript presented in an intelligible fashion and written in standard English?

Reviewer #1: Yes

Reviewer #2: Yes

5. Review Comments to the Author

Reviewer #1: Wise et al have investigated the impact of elevated intra-abdominal pressure (IAP) on physiological variables in two different experimental models in pigs. The study represents original research describing in details the changes in cardiovascular, respiratory, renal, gastrointestinal and hematological function and metabolism response elevated IAP. Albeit the same group or other authors have previously reported similar findings, the manuscript is comprehensive summary of the pathophysiological alterations that may be observed in experimental setting of intra-abdominal hypertension (IAH). As such, it is valuable information for researchers in the field. As an additional value, the authors compare the impact of two different models, pneumoperitoneum and intestinal obstruction model, respectively.

The manuscript contains extensive amount of data. My first main concern is the way the data are presented. The main aim of the study remains unclear, and this makes the manuscript vague. Was the main aim to describe the physiological changes in response to increased IAP or was it to compare the two models of IAH? If it was the latter (as it noted in introduction and in title), then the data should be presented accordingly, demonstrating the presence or absence of differences between these two as a first priority. At present, Tables 1 to 5 refer to physiological variables regardless of IAP model, while the data of two separate models are find in supplement. This is inconsistent with study title and hypothesis.

My second main concern is the length of discussion. With 8 pages (!) in total it resembles PhD thesis rather than single scientific report. The discussion should shortened considerably, to max of 4 pages, with reduction of repeated description of your own findings, and replacing this with discussion on comparative findings from other relevant studies.

Other comments:

Abstract:

- Prolonged intensive care and hospital stay are not relevant in context of present study. Renal dysfunction, morbidity, multi-organ failure are unnecessary repetition.

- Please rephrase the results section. Choose the most relevant indices, report their numerical values.

- Conclusions: last sentence is speculative, not precisely in context of present study. Consider rephrasing.

Introduction

- Many of the references 1-10 are out of date. More recent data are available. For example Reintam Blaser A, et al. Crit Care Med. 2019; 47(4):535-542

- Paragraph on polycompartment syndrome is not necessary. Instead, please open more your hypothesis. Why the two models of IAP should have different impact on physiological variables and organ function? Elaborate on the inflammatory response to mechanical obstruction. Is it proven in pig model?

Study design

- As I understand, the animals were immediately in the start of experiment exposed to either 20, 30 or 40 mmHg of IAP. One may argue that such high pressures (especially of 30 and 40 mmHg) very unlikely would develop quickly, in minutes. Rather, the gradual increase to 20, and then to 30, and 40 mmHg would more to resemble the real life clinical scenario. Please explain your considerations for choosing the study design.

- Please elaborate the similarities and differences between pigs and humans in respect to IAP-s. Is 20 mmHg in pigs’ similar severity as in humans? Forty mmHg would be rather exceptional in humans, quickly leading to an catastrophic deterioration. What is the rationale to test this value in pig model?

- Page 9, first paragraph. WSACS guidelines do not describe the bladder pressure measurements in experimental animals. Please correct.

Statistical analysis

- What was the primary endpoint for sample size calculation? Please indicate it in main text, not in supplementary materials.

- Please confirm the data in Tables 1 to 5 are all normally distributed, and mean (SD) is correct way of data presentation. Please avoid using ± in mean (SD) presentation. Please check the decimals after comma, unify through the Tables.

- Pig-specific variances were allowed (Page 11, first paragraph). Please explain what that means.

- This is secondary, but more complete analysis, of a study previously published by the same group (16). How much of data exactly have been published previously? How many experiments out of 49 animals were included in the previous report?

Results

- Restructure the data presentation according to the main hypothesis

- Calculated glomerular filtration. What you mean with: “Analysis of calculated glomerular filtration was performed using data collected from transvesical, transperitoneal, and transgastric measures.“ Please explain, consider rephrasing.

Discussion

- The first sentence: This study quantified the impact of two different models on physiological parameters in pigs. This is vague. Above all, the results in present form (the Tables and Figures, which attract the first attention) demonstrate the effect of IAP on organ systems, regardless of the IAP model. Either restructure the results, or change the introduction and discussion.

- For most of cardiovascular indices, no differences between the models were observed. What that shows? I am not sure we should expect the differences here; I am in doubt with the hypothesis that these models have different impact on physiology. In opposite, I would put forward the hypothesis that the impact is similar for both models.

- The discussion is far too detailed in many aspects. This is not necessary, please shorten, generalize, underline the most important findings and compare them with existing knowledge.

- In your preliminary study (ref 16) it is concluded that the most relevant parameters to evaluate the deleterious effects of IAH are monitoring of APP, Cdyn, pHi and lactate. Does your current study confirms the preliminary data? Please discuss.

- Please avoid repeated presentation of results!

Conclusions

- Second sentence: „Using SVV or PPV in the presence of IAH is an unreliable way of assessing volume status.” How did you come to this? The volume status (fluid responsiveness, for example) was not specifically assessed; these indices were not compared to others in that respect. Please stick precisely only on observed findings while making the conclusions.

Reviewer #2: The authors studied the pathophysiological effects of elevated IAP in animal models of pneumoperitoneum (Pn) and mechanical intestinal obstruction (MIO) as part of a complex, exploratory animal study.

Based on the experiments conducted and the results presented, the authors have performed a very comprehensive, detailed and thorough comparative data analysis with a high level of statistical support. The value of the study is that the observed changes are discussed by organ system.

However, the scientific value of this huge study, the ability to pick out the truly relevant pathophysiological variations is not an easy task for the reader. Therefore, the reviewer's main suggestion is that, instead of a meaningless and general conclusion, either in the Conclusion or in the Potential clinical implications chapter, the authors should emphatically summarise the important differences found in the two models when comparing the damage to different organ systems.

Additional comments:

1. The element numbers (n) reported in the Study design description on page 8 do not match the element number data reported in the first row of Tables 1-5 of the Results.

2. Why Table 1-5 only shows the data for the Pn groups and the significance value detected compared to the control group. Why do we not see the data and statistical differences of the MIO groups compared to the control or PN groups?

3. In line 4 of the Study design, the information reads "Each group was divided in half and studied for three and five hours". This raises the question of whether the tables present data for the 3 or 5 hour study?

4. On page 26, in the Limitations, do not refer to "insufficient grant money" when there were deaths of pigs in several study groups during the trial (Figures S2, S7, S11, S12).

5. The 5-hour study is sufficient to analyse correlations between IAP and the other parameters tested. However, the 5 h is not sufficient to explain the pathophysiological difference between the two models in terms of inflammatory responses, as hypothesized. This would have required longer study time and the determination of more inflammatory markers.

6. PLOS authors have the option to publish the peer review history of their article (what does this mean?). If published, this will include your full peer review and any attached files.

Reviewer #1: No

Reviewer #2: No

---

## [Author Response · Author response to Decision Letter 0]

23 May 2023

Dear Editor of PLOS One

We are grateful for the comprehensive review with advice and suggestions from the reviewers. We have amended the manuscript according to their reviews, with significant changes to the clarity of the introduction and objectives, presentation of the results, and shortening of the discussion (focussing on the primary objective). We have tried to explain those statistical areas and design of the study that wasn’t clear. We have made corrections as advised by the reviewers and attempted to answer all of their questions below.

Thank you for your time to look at the manuscript again and we look forward to hearing back.

Reviewer #1: 

Wise et al have investigated the impact of elevated intra-abdominal pressure (IAP) on physiological variables in two different experimental models in pigs. The study represents original research describing in detail the changes in cardiovascular, respiratory, renal, gastrointestinal and hematological function and metabolism response elevated IAP. Albeit the same group or other authors have previously reported similar findings, the manuscript is comprehensive summary of the pathophysiological alterations that may be observed in experimental setting of intra-abdominal hypertension (IAH). As such, it is valuable information for researchers in the field. As an additional value, the authors compare the impact of two different models, pneumoperitoneum and intestinal obstruction model, respectively.

The manuscript contains extensive amount of data. My first main concern is the way the data are presented. 

1. The main aim of the study remains unclear, and this makes the manuscript vague. Was the main aim to describe the physiological changes in response to increased IAP or was it to compare the two models of IAH? If it was the latter (as it noted in introduction and in title), then the data should be presented accordingly, demonstrating the presence or absence of differences between these two as a first priority. 

We understand that our aims were not stated clearly enough. We thank the reviewer for highlighting this and have altered the text to better describe what our objectives were. The primary aim was a description of the physiological effects of IAP, and the secondary objective was a comparison of these effects between the 2 models used against a control group. We have amended the text to make it clearer to the reader.

The amended text reads as follows:

“The primary aim was to study the systemic effects of raised IAP on various organ systems. We used both the pneumoperitoneum and mechanical obstruction models to generate elevated intra-abdominal pressure. Data from all pigs, regardless of the model used, were included in the analysis for the primary outcome. We felt it important to study these effects across various IAP levels, as this has not been well documented previously. The secondary aim was to describe the effects between the two different models. We hypothesised that there would be no conceivable difference between the two models because of the relatively short duration of raised IAP. 

2. At present, Tables 1 to 5 refer to physiological variables regardless of IAP model, while the data of two separate models are find in supplement. This is inconsistent with study title and hypothesis.

We agree with the reviewer, and we have changed the title to better fit our objectives:

“The pathophysiological impact of intra-abdominal hypertension in pigs” 

3. My second main concern is the length of discussion. With 8 pages (!) in total it resembles PhD thesis rather than single scientific report. The discussion should shorten considerably, to max of 4 pages, with reduction of repeated description of your own findings, and replacing this with discussion on comparative findings from other relevant studies.

We have shortened the discussion considerably and removed repeated findings. We have focussed on the primary objective with reference to the secondary objective findings in the supplementary material, otherwise there is simply too much information. 

Other comments:

Abstract:

4. Prolonged intensive care and hospital stay are not relevant in context of present study. Renal dysfunction, morbidity, multi-organ failure is unnecessary repetition.

Thank you for this suggestion, we have amended the text accordingly:

“Intra-abdominal hypertension and abdominal compartment syndrome are common with clinically significant consequences.”

5. Please rephrase the results section. Choose the most relevant indices, report their numerical values.

We agree that this section required clarity and have made changes as per the reviewer’s request.

“There were significant changes (p<0.05) noted in all organ systems, most notably systolic blood pressure (SBP) (p<0.001), cardiac index (CI) (p=0.003), stroke volume index (SVI) (p<0.001), mean pulmonary airway pressure (MPP) (p<0.001), compliance (p<0.001), pO2 (p=0.003), bicarbonate (p=0.041), hemoglobin (p=0.012), lipase (p=0.041), total bilirubin (p=0.041), gastric pH (p<0.001), calculated glomerular filtration rate (GFR) (p<0.001), and urine output (p<0.001). SVV increased progressively as the IAP increased with no obvious changes in intravascular volume status. When describing the two models, there were no significant differences between the models regarding their impact on cardiovascular, respiratory, renal and gastrointestinal systems. However, significant differences were noted between the two models at 30mmHg, with MIO30 showing worse metabolic and hematological parameters, and Pn30 and Pn40 showing a more rapid rise in creatinine.” 

6. Conclusions: last sentence is speculative, not precisely in context of present study. Consider rephrasing.

We agree and thank the reviewer for the suggestion and have amended the text in accordance with the comment.

“This study identified and quantified the impact of intra-abdominal hypertension at different pressures on several organ systems and highlighted the significance of even short-lived elevations. Two models of intra-abdominal pressure were used, with a mechanical obstruction model showing more rapid changes in metabolic and haematological changes. Differences between the two models may represent different underlying cellular and vascular pathophysiological processes, but this remains unclear”. 

Introduction

7. Many of the references 1-10 are out of date. More recent data are available. For example, Reintam Blaser A, et al. Crit Care Med. 2019; 47(4):535-542

Thank you for the suggestion, we have updated the references, accordingly, having removed those that were published before 2010.

Smit, M., Koopman, B., Dieperink, W. et al. Intra-abdominal hypertension and abdominal compartment syndrome in patients admitted to the ICU. Ann. Intensive Care 10, 130 (2020). https://doi.org/10.1186/s13613-020-00746-9

Reintam Blaser, Annika MD, PhD1,2; Regli, Adrian MD3,4,5; De Keulenaer, Bart MD3,6; Kimball, Edward J. MD7; Starkopf, Liis MSc8; Davis, Wendy A. MPH, PhD4; Greiffenstein, Patrick MD, FACS9; Starkopf, Joel MD, PhD1,10; the Incidence, Risk Factors, and Outcomes of Intra-Abdominal (IROI) Study Investigators. Incidence, Risk Factors, and Outcomes of Intra-Abdominal Hypertension in Critically Ill Patients—A Prospective Multicenter Study (IROI Study). Critical Care Medicine 47(4):p 535-542, April 2019. | DOI: 10.1097/CCM.0000000000003623

Muturi, A., Ndaguatha, P., Ojuka, D. et al. Prevalence and predictors of intra-abdominal hypertension and compartment syndrome in surgical patients in critical care units at Kenyatta National Hospital. BMC Emerg Med 17, 10 (2016). https://doi.org/10.1186/s12873-017-0120-y

8. Paragraph on polycompartment syndrome is not necessary. Instead, please open more your hypothesis. Why the two models of IAP should have different impact on physiological variables and organ function? Elaborate on the inflammatory response to mechanical obstruction. Is it proven in pig model?

We are grateful for this suggestion and have amended the text as suggested, discussing possible effects on gastrointestinal vasculature.

“The interaction between different organ systems within neighbouring anatomical compartments has been described as polycompartment syndrome (14, 15). It is for this reason we have studied multiple organ systems together. This study includes two models of raised intraabdominal pressure, a pneumoperitoneum and a mechanical intestinal obstruction model (16). We have previously demonstrated that intestinal vascular occlusion may differ between a pneumoperitoneum versus a mechanically obstructed model. This may affect organ systems differently. Lactate increased substantially in a mechanically obstructed model which may reflect increased anaerobic metabolism because of imbalances in cardiorespiratory dynamics. This may in turn promote an inflammatory state but has yet to be proven, and probably requires a longer period of sustained elevated IAP.”

Párraga Ros E, Correa-Martín L, Sánchez-Margallo FM, Candanosa-Aranda IE, Malbrain MLNG, Wise R, Latorre R, López Albors O, Castellanos G. Intestinal histopathological changes in a porcine model of pneumoperitoneum-induced intra-abdominal hypertension. Surg Endosc. 2018 Sep;32(9):3989-4002. doi: 10.1007/s00464-018-6142-z. Epub 2018 May 17. PMID: 29777353.

Correa-Martín L, Párraga E, Sánchez-Margallo FM, Latorre R, López-Albors O, Wise R et al (2016) Mechanical intestinal obstruction in a porcine model: effects of intra-abdominal hyper- tension. A preliminary study. PLoS ONE 11(2):e0148058 

Study design

9. As I understand, the animals were immediately in the start of experiment exposed to either 20, 30 or 40 mmHg of IAP. One may argue that such high pressures (especially of 30 and 40 mmHg) very unlikely would develop quickly, in minutes. Rather, the gradual increase to 20, and then to 30, and 40 mmHg would more to resemble the real-life clinical scenario. Please explain your considerations for choosing the study design.

We agree with the reviewer’s comment, and a graduated increase in pressure would have simulated a closer to life scenario. However, we chose to study the pigs at separate pressures maintained for the specified time periods because of concerns that many of the pigs may not have survived to reach 40mmHg and we would then not have been able to observe what happens at this pressure. Also, the response of the pigs to pressures for so long would have been difficult to predict and resulted in the size of the groups changing as pigs died. Under these study conditions we were also able to study extreme pressures, like 40mmHg, which would not be possible in most clinical settings. We have added a comment to the limitations section in response to this useful suggestion.

“Set pressures were used versus gradually increasing pressures over time which may have been closer to clinical situations. However, we chose this design to define the size of the groups, as pigs may not have survived raised pressures for long periods, thus missing out on observing what happens at 30mmHg and 40mmHg.”

10. Please elaborate the similarities and differences between pigs and humans in respect to IAP-s. Is 20 mmHg in pigs’ similar severity as in humans? Forty mmHg would be rather exceptional in humans, quickly leading to a catastrophic deterioration. What is the rationale to test this value in pig model?

There are little data looking specifically at similarities between human and pigs with respect to intra-abdominal pressure. Our assumption is based on several previous studies that have suggested the utility in using pigs for the study of anatomical and pathological process due to the similarity in anatomical size and structure, physiology, and immunology. All animal models have inherent limitations, and we would not expect a 1 to 1 correlation from any animal model to human. As we have noted in our previous response, the use of these pressure allows unique insight into the physiological effects of these high pressures. 

Very high IAPs are rarely studied. We decided it would be useful to study both commonly encountered pressures, as well as extremes in pressures, as this may highlight pathological processes not easily identified unless sustained at lower pressures for very long periods, which would be difficult to simulate in a laboratory.

Lunney JK, Van Goor A, Walker KE, Hailstock T, Franklin J, Dai C. Importance of the pig as a human biomedical model. Sci Transl Med. 2021 Nov 24;13(621):eabd5758. doi: 10.1126/scitranslmed.abd5758. Epub 2021 Nov 24. PMID: 34818055.

Hou N, Du X, Wu S. Advances in pig models of human diseases. Animal Model Exp Med. 2022 Apr;5(2):141-152. doi: 10.1002/ame2.12223. Epub 2022 Mar 27. PMID: 35343091; PMCID: PMC9043727.

11. Page 9, first paragraph. WSACS guidelines do not describe the bladder pressure measurements in experimental animals. Please correct.

Thank you, we realise this needed clarification. We worded it in this way to describe the method (and not have to go into detail) but have highlighted that this technique is the same gold standard technique from guidelines for humans.

“Gastric and bladder pressures were measured using the same methods as those described for humans in the WSACS consensus guidelines”

Statistical analysis

12. What was the primary endpoint for sample size calculation? Please indicate it in main text, not in supplementary materials.

Thank you, we have made the adjustment as you have suggested.

“We performed a statistical power analysis for sample size estimation from data analysing hemodynamic parameters from previous pig studies with an alpha of 0.05 and power equal to 0.80.”

13. Please confirm the data in Tables 1 to 5 are all normally distributed and mean (SD) is correct way of data presentation. Please avoid using ± in mean (SD) presentation. Please check the decimals after comma, unify through the Tables.

We have made the changes to the tables as requested with standard deviations in brackets, and the title of each table indicating that the mean is presented. 

We can confirm that the data is normally distributed, and we have checked the correct decimal placement.

14. Pig-specific variances were allowed (Page 11, first paragraph). Please explain what that means.

The statistical analysis assumed that not all pigs were physiologically the same. This was accounted for in the linear mixed model.

For example, a LMM could include terms to account for variation within each pig measurements. Consider the model

Y = b0 + b1x1 + b2x2 + a1z1 + a2z2 + cisigmai^2 + some error

The terms x1 and x2 are fixed effects

The terms z1 and z2 are random effects

And they account for the pigs variation in sigmai^2.

15. This is secondary, but more complete analysis, of a study previously published by the same group (16). How much of data exactly have been published previously? How many experiments out of 49 animals were included in the previous report?

The initial publication was a much smaller sample from the initial experiment with only 15 pigs included, all of them in the mechanical obstruction model and only looking at 20mmHg pressure at 2 and 5 hours. 

Results

16. Restructure the data presentation according to the main hypothesis

We agree with the reviewer in that the primary and secondary objectives were not clear enough to match the results. We also agree with other comments that the results section needs to be clearer and simplified for the reader. As such, we have focussed the results section on the primary objective findings. We have included a section after reporting the primary findings at the end of the results section as a short synopsis of the secondary objectives, with reference to the supplementary data with the reader wishes to delve into those findings. We have restructured and make extensive changes, hopefully meeting the expectations of the reviewers.

17. Calculated glomerular filtration. What you mean with: “Analysis of calculated glomerular filtration was performed using data collected from transvesical, transperitoneal, and transgastric measures. “ Please explain, consider rephrasing.

Glomerular filtration (not glomerular filtration rate) was calculated from MAP and IAP (Filtration gradient = MAP – [2 x IAP]). Estimation of GF is typical in porcine models where specific renal drug excretion (e.g., chromium-51 or technetium-99m) is not possible to measure a true GFR. At the time of the study design, it was felt that the short duration of the study may have led to an unpredictable measure of creatinine clearance. 

We have changed the phrasing in the text to read as follows: Glomerular filtration was calculated using the formula: filtration gradient = MAP – (2 x IAP). IAP measures were obtained transvesically, transperitoneally, and transgastrically.

Discussion 

18. The first sentence: This study quantified the impact of two different models on physiological parameters in pigs. This is vague. Above all, the results in present form (the Tables and Figures, which attract the first attention) demonstrate the effect of IAP on organ systems, regardless of the IAP model. Either restructure the results or change the introduction and discussion.

We would like to thank the reviewer for this advice. We have changed the title, restructured the results, and focused the introduction and discussion on the primary objective, which is hopefully clearer for the reader.

19. For most of cardiovascular indices, no differences between the models were observed. What that shows? I am not sure we should expect the differences here; I am in doubt with the hypothesis that these models have different impact on physiology. In opposite, I would put forward the hypothesis that the impact is similar for both models.

Thank you, we agree that we did not make our thoughts about this clear enough and have amended the manuscript accordingly. We agree with the reviewer’s thoughts and rephrased the hypothesis so that it is more clearly and links directly to our intended message.

20. The discussion is far too detailed in many aspects. This is not necessary, please shorten, generalize, underline the most important findings and compare them with existing knowledge.

Thank you for the suggestion. We have tried to do this with significant shortening of the discussion, focussing on salient points related mostly to the primary objective, and comparisons with published literature.

21. In your preliminary study (ref 16) it is concluded that the most relevant parameters to evaluate the deleterious effects of IAH are monitoring of APP, Cdyn, pHi and lactate. Does your current study confirm the preliminary data? Please discuss.

APP has previously been touted as a useful predictor of IAH injury and an early warning sign to hasten therapeutic intervention. However, our data showed several non-normally distributed data. This may have been because of the “mast-suit” type effect from early raised IAP in some pigs that tended to make it an unreliable marker and difficult to analyse and interpret. Added to this is the clinical scenario of the additional sue of inotropes which can create a higher APP, but where splanchnic perfusion may be much worse. As a result, we dd not report APP.

Dynamic compliance, pHi and lactate all remain relevant parameters in evaluating the deleterious effects of IAH. However, several other parameters from each organ system appear just as useful, including SVI, mean pulmonary pressure, driving pressure, pCO2 gap, and urine output. 

However, this study did not aim to confirm this aspect of the first analysis, but rather to first describe the changes seen in the animals at the different pressure levels. We have not compared the utility of different parameters as this would further complicate the paper. 

22. Please avoid repeated presentation of results!

Thank you for highlighting this error, we have corrected where needed.

Conclusions

23. Second sentence: „Using SVV or PPV in the presence of IAH is an unreliable way of assessing volume status.” How did you come to this? The volume status (fluid responsiveness, for example) was not specifically assessed; these indices were not compared to others in that respect. Please stick precisely only on observed findings while making the conclusions.

We thank the reviewer for this observation and have removed this statement from the conclusion and amended our discussion. Our findings support previous studies as opposed to directly identifying SVV and PVV as being an unreliable way of assessing volume status. Changes in SVV and PVV are supposed to predict volume status. In our model, animals’ volume status was not directly assessed, but equally, they did not lose any fluid and were given IV fluid throughout the study. Interstitial fluid loss was not accounted for hence we agree with the reviewer that this cannot be presented as a direct finding, but rather in support of previous studies which have found SVV and PPVV to be inaccurate in the cases of raised IAP.

Duperret S, Lhuillier F, Piriou V, Vivier E, Metton O, Branche P, et al. Increased intra-abdominal pressure affects respiratory variations in arterial pressure in normovolaemic and hypovolaemic mechanically ventilated healthy pigs. Intensive Care Med. 2007; 33(1):163–71. PMID: 17102964 

Renner J, Gruenewald M, Quaden R, Hanss R, Meybohm P, Steinfath M, et al. Influence of increased intra-abdominal pressure on fluid responsiveness predicted by pulse pressure variation and stroke volume variation in a porcine model. Crit Care Med. 2009; 37(2):650–8. doi: 10.1097/CCM. 0b013e3181959864 PMID: 19114894 

Jacques D, Bendjelid K, Duperret S, Colling J, Piriou V, Viale JP. Pulse pressure variation and stroke volume variation during increased intra-abdominal pressure: an experimental study. Crit Care. 2011; 15(1):R33. doi: 10.1186/cc9980 PMID: 21247472 

Reviewer #2: 

The authors studied the pathophysiological effects of elevated IAP in animal models of pneumoperitoneum (Pn) and mechanical intestinal obstruction (MIO) as part of a complex, exploratory animal study.

Based on the experiments conducted and the results presented, the authors have performed a very comprehensive, detailed and thorough comparative data analysis with a high level of statistical support. The value of the study is that the observed changes are discussed by organ system.

However, the scientific value of this huge study, the ability to pick out the truly relevant pathophysiological variations is not an easy task for the reader. Therefore, the reviewer's main suggestion is that, instead of a meaningless and general conclusion, either in the Conclusion or in the Potential clinical implications chapter, the authors should emphatically summarise the important differences found in the two models when comparing the damage to different organ systems.

We are grateful for these comments and suggestions from the reviewer. We have worked on the discussion and made significant changes and hope that it now meets expectations. Our primary aim was to describe the pathophysiological changes, with the comparison of the 2 models being the secondary aim. We have tried to make this clearer.

Additional comments:

1. The element numbers (n) reported in the Study design description on page 8 do not match the element number data reported in the first row of Tables 1-5 of the Results.

We thank the reviewer for highlighting this section as being unclear. There were 49 pigs in total. 

Control = 5

Pneumoperitoneum 20mmHg = 10

Pneumoperitoneum 30mmHg = 10

Pneumoperitoneum 40mmHg = 10

Mechanical Intestinal Obstruction 20mmHg = 9

Mechanical Intestinal Obstruction 30mmHg = 5

Making the following numbers for the table:

Control: = 5

IAP 20mmHg = 19

IAP 30mmHg = 15

IAP 40mmHg = 10

The swine were consecutively allocated into six groups: a single control group (Cr; n=5), three pneumoperitoneum groups with IAPs of 20 mmHg (Pn20; n=10), 30 mmHg (Pn30; n=10), and 40 mmHg (Pn40; n=10), and two mechanical intestinal occlusion groups with IAPs of 20 mmHg (MIO20; n=9) and 30 mmHg (MIO30; n=5).

2. Why Table 1-5 only shows the data for the Pn groups, and the significance value detected compared to the control group. Why do we not see the data and statistical differences of the MIO groups compared to the control or PN groups?

We understand that this needed to be clearer. We have renamed the tables, explained the primary and secondary objectives more clearly, and reworded the start of the results section to hopefully bring clarity. The tables represent the pooled data from all groups regardless of IAP model and reflect our attempt to describe our primary objective, namely the physiological effects of raised IAP on the various organ systems. Hence the number of subjects as described above. 

3. In line 4 of the Study design, the information reads "Each group was divided in half and studied for three and five hours". This raises the question of whether the tables present data for the 3- or 5-hour study?

We thank the reviewer for this comment and have amended the wording so as not to cause confusion.

All pigs were included in the analysis for the primary objective. The assignment of pigs to the different treatments, combining condition, pressure and duration, were as follows: Not all combinations were present, implying “missingness” by design. Contrasts are therefore required to make the appropriate comparisons. Such comparisons typically only make use of a subset of the data. NOTE: It was assumed, that the duration conditions are completely equivalent except for those measurements at the end. In other words, it is assumed that there was no reason to expect that the evolution ran differently at the earlier stages, and the additional measurements in the longer duration only showed how the evolution continues. It was thus possible to compare the evolution for each physiological parameter and how it may differ depending on the model and the pressure. The control condition with 0 pressure compared to the obstruction and pneumoperitoneum conditions with increasing pressure (40mmHg only for pneumoperitoneum). The obstruction and pneumoperitoneum compared directly with a pressure of 20mmHg or 30mmHg. 

We have attached the statical analysis for further review if required, but after consultation with the biostatistician, are confident of the linear mixed model created for analysis.

4. On page 26, in the Limitations, do not refer to "insufficient grant money" when there were deaths of pigs in several study groups during the trial (Figures S2, S7, S11, S12).

Thank you for this comment, which we have addressed. The deaths in those groups were considered as outcomes, however, the reviewer is correct in saying that we were not able to make the groups the same size because for the shortfall. However, we did meet appropriate sample size. We have reworded this sentence and are happy to change it again if recommended otherwise.

“Due to funding limitations, one group had four, which still satisfied our sample size requirements.”

5. The 5-hour study is sufficient to analyse correlations between IAP and the other parameters tested. However, the 5 h is not sufficient to explain the pathophysiological difference between the two models in terms of inflammatory responses, as hypothesized. This would have required longer study time and the determination of more inflammatory markers.

Once again, we hank the reviewer for this comment and suggestion and have explored these concepts in the discussion.

---

## [Decision Letter · Decision Letter 1]

4 Jul 2023

PONE-D-22-32095R1The pathophysiological impact of intra-abdominal hypertension in pigsPLOS ONE

Dear Dr. Wise,

Thank you for submitting your manuscript to PLOS ONE. After careful consideration, we feel that it has merit but does not fully meet PLOS ONE’s publication criteria as it currently stands. Therefore, we invite you to submit a revised version of the manuscript that addresses the points raised during the review process. Please submit your revised manuscript by Aug 18 2023 11:59PM. If you will need more time than this to complete your revisions, please reply to this message or contact the journal office at plosone@plos.org. Please include the following items when submitting your revised manuscript:A rebuttal letter that responds to each point raised by the academic editor and reviewer(s). You should upload this letter as a separate file labeled 'Response to Reviewers'.A marked-up copy of your manuscript that highlights changes made to the original version. You should upload this as a separate file labeled 'Revised Manuscript with Track Changes'.An unmarked version of your revised paper without tracked changes. You should upload this as a separate file labeled 'Manuscript'.If applicable, we recommend that you deposit your laboratory protocols in protocols.io to enhance the reproducibility of your results. Protocols.io assigns your protocol its own identifier (DOI) so that it can be cited independently in the future. For instructions see: https://journals.plos.org/plosone/s/submission-guidelines#loc-laboratory-protocols. Additionally, PLOS ONE offers an option for publishing peer-reviewed Lab Protocol articles, which describe protocols hosted on protocols.io. Read more information on sharing protocols at https://plos.org/protocols?utm_medium=editorial-email&utm_source=authorletters&utm_campaign=protocols.

We look forward to receiving your revised manuscript.

Kind regards,

Alexander Wolf

Academic Editor

PLOS ONE

Journal Requirements:

**Additional Editor Comments:**

Thank you for your revised manuscript, which has improved significantly. However, there are still objections from a reviewer regarding the statistics, which you should addres.

In this regard, I noticed that you use a one-way ANOVA for normal distribution. As a logical consequence, a Kruskal Wallis test should be done when the distribution is not normal.

I recommend to combine the one-way ANOVA and the Kruskal Wallis test in case of more than two groups to be compared with an appropriate post-hoc test.

Reviewers' comments:

Reviewer's Responses to Questions

**Comments to the Author**

1. If the authors have adequately addressed your comments raised in a previous round of review and you feel that this manuscript is now acceptable for publication, you may indicate that here to bypass the “Comments to the Author” section, enter your conflict of interest statement in the “Confidential to Editor” section, and submit your "Accept" recommendation.

Reviewer #1: All comments have been addressed

Reviewer #2: All comments have been addressed

2. Is the manuscript technically sound, and do the data support the conclusions?

Reviewer #1: Yes

Reviewer #2: Yes

3. Has the statistical analysis been performed appropriately and rigorously? 

Reviewer #1: Yes

Reviewer #2: Yes

4. Have the authors made all data underlying the findings in their manuscript fully available?

Reviewer #1: Yes

Reviewer #2: Yes

5. Is the manuscript presented in an intelligible fashion and written in standard English?

Reviewer #1: Yes

Reviewer #2: Yes

6. Review Comments to the Author

Reviewer #1: The revion has siginficantly improved the manuscript. I found my critique adequately adressed. I have only one conern with respect of data presentation:

Authors describe in Statistical analysis section, that continuous variables are presented as means with standard deviations when normally distributed, and as medians with interquartile ranges when non-normal distribution occurred.

In Tables, all data are presented means(SD). Please confirm that all data are indeed normally distributed. With biological data with relatively low number of experiments per group, this would be unusual. Please correct, if needed. If some data are not normally distributed, then entire Table should be presented as medians and IQR.

Reviewer #2: The authors made significant changes to the manuscript text, thus, the quality of the study reached an adequate standard.

7. PLOS authors have the option to publish the peer review history of their article (what does this mean?). If published, this will include your full peer review and any attached files.

Reviewer #1: No

Reviewer #2: No

---

## [Author Response · Author response to Decision Letter 1]

19 Jul 2023

Dear Editor

Reviewer #1:

Question 1:

Please find our responses to the two remaining questions, the first from reviewer #1, and the second a comment and suggestion from the editor. The revision has significantly improved the manuscript. I found my critique adequately addressed. I have only one concern with respect of data presentation: Authors describe in Statistical analysis section, that continuous variables are presented as means with standard deviations when normally distributed, and as medians with interquartile ranges when non-normal distribution occurred. In Tables, all data are presented means (SD). Please confirm that all data are indeed normally distributed. With biological data with relatively low number of experiments per group, this would be unusual. Please correct, if needed. If some data are not normally distributed, then entire Table should be presented as medians and IQR.

Response: Thank you for this comment and correction of the analysis. Almost all data were not normally distributed. The tables have been corrected with medians and inter-quartile ranges.

Additional Editor Comments:

Question 2:

Thank you for your revised manuscript, which has improved significantly. However, there are still objections from a reviewer regarding the statistics, which you should address. In this regard, I noticed that you use a one-way ANOVA for normal distribution. As a logical consequence, a Kruskal Wallis test should be done when the distribution is not normal. I recommend to combine the one-way ANOVA and the Kruskal Wallis test in case of more than two groups to be compared with an appropriate post-hoc test.

Response: As the data were not normally distributed, we have corrected the tables and presented the data as median and inter-quartile ranges. We have repeated the analysis using a Kruskal Wallis test and corrected the p-values. The slight changes in the p values have not changed the findings. We have added the following to the text:

“... Normality was evaluated with the Kolmogorov-Smirnov test. One-way ANOVA or Kruskal-Wallis tests were used to compare conAnuous variables as appropriate.”

Kind regards

---

## [Decision Letter · Decision Letter 2]

9 Aug 2023

The pathophysiological impact of intra-abdominal hypertension in pigs

PONE-D-22-32095R2

Dear Dr. Rober Wise,

We’re pleased to inform you that your manuscript has been judged scientifically suitable for publication and will be formally accepted for publication once it meets all outstanding technical requirements.

Kind regards,

Nataša Kovač, MD, PhD

Academic Editor

PLOS ONE

Additional Editor Comments (optional):

Reviewers' comments:

Reviewer's Responses to Questions

**Comments to the Author**

1. If the authors have adequately addressed your comments raised in a previous round of review and you feel that this manuscript is now acceptable for publication, you may indicate that here to bypass the “Comments to the Author” section, enter your conflict of interest statement in the “Confidential to Editor” section, and submit your "Accept" recommendation.

Reviewer #1: All comments have been addressed

Reviewer #2: All comments have been addressed

Reviewer #3: (No Response)

2. Is the manuscript technically sound, and do the data support the conclusions?

Reviewer #1: Yes

Reviewer #2: Yes

Reviewer #3: Yes

3. Has the statistical analysis been performed appropriately and rigorously? 

Reviewer #1: Yes

Reviewer #2: Yes

Reviewer #3: I Don't Know

4. Have the authors made all data underlying the findings in their manuscript fully available?

Reviewer #1: (No Response)

Reviewer #2: Yes

Reviewer #3: Yes

5. Is the manuscript presented in an intelligible fashion and written in standard English?

Reviewer #1: Yes

Reviewer #2: Yes

Reviewer #3: Yes

6. Review Comments to the Author

Reviewer #1: (No Response)

Reviewer #2: The authors have addressed most of the reviewers comments by adding new data and/or rewriting the manuscript. Therefore, I do not have further comments.

Reviewer #3: The authors have stated: "A statistical power analysis was performed for sample size estimation, based on data from previous animal studies [1-5]. With an alpha of 0.05

and power = 0.80, the projected sample size needed with this effect size for the endpoint of heart rate was 2. This is based on a starting heart rate

of 105 ± 10 with a predicted increase of at least 20%."

Could you just provide additional info on other parameters used for the calculation of sample size, for example, which R package and which method? Was it some method for mixed effects model sample size calculation?

7. PLOS authors have the option to publish the peer review history of their article (what does this mean?). If published, this will include your full peer review and any attached files.

Reviewer #1: No

Reviewer #2: No

Reviewer #3: No

---

## [Editor Report · Acceptance letter]

16 Aug 2023

PONE-D-22-32095R2 

The pathophysiological impact of intra-abdominal hypertension in pigs  

Dear Dr. Wise:

I'm pleased to inform you that your manuscript has been deemed suitable for publication in PLOS ONE. Congratulations! Your manuscript is now with our production department. 

Kind regards, 

on behalf of

Dr. Nataša Kovač 

Academic Editor

PLOS ONE